# Distributed Bayesian Posterior Sampling via Moment Sharing

**Minjie Xu[1]**[*]**, Balaji Lakshminarayanan[2], Yee Whye Teh[3], Jun Zhu[1], and Bo Zhang[1]**

[1]State Key Lab of Intelligent Technology and Systems; Tsinghua National TNList Lab
[1]Department of Computer Science and Technology, Tsinghua University, Beijing 100084, China
[2]Gatsby Unit, University College London, 17 Queen Square, London WC1N 3AR, UK
[3]Department of Statistics, University of Oxford, 1 South Parks Road, Oxford OX1 3TG, UK

## Abstract

We propose a distributed Markov chain Monte Carlo (MCMC) inference algorithm for large scale Bayesian posterior simulation. We assume that the dataset is partitioned and stored across nodes of a cluster. Our procedure involves an independent MCMC posterior sampler at each node based on its local partition of the data. Moment statistics of the local posteriors are collected from each sampler and propagated across the cluster using expectation propagation message passing with low communication costs. The moment sharing scheme improves posterior estimation quality by enforcing agreement among the samplers. We demonstrate the speed and inference quality of our method with empirical studies on Bayesian logistic regression and sparse linear regression with a spike-and-slab prior.

## 1 Introduction

As we enter the age of "big data", datasets are growing to ever increasing sizes and there is an urgent need for scalable machine learning algorithms. In Bayesian learning, the central object of interest is the posterior distribution, and a variety of variational and Markov chain Monte Carlo (MCMC) methods have been developed for "big data" settings. The main difficulty with both approaches is that each iteration of these algorithms requires an impractical $O(N)$ computation for a dataset of size $N \gg 1$. There are two general solutions: either to use stochastic approximation techniques based on small mini-batches of data [15, 4, 5, 20, 1, 14], or to distribute data as well as computation across a parallel computing architecture, e.g. using MapReduce [3, 13, 16].

In this paper we consider methods for distributing MCMC sampling across a computer cluster where a dataset has been partitioned and locally stored on the nodes. Recent years have seen a flurry of research on this topic, with many papers based around "embarrassingly parallel" architectures [16, 12, 19, 9]. The basic thesis is that because communication costs are so high, it is better for each node to run a separate MCMC sampler based on its data stored locally, completely independently from others, and then for a final combination stage to transform the local samples into samples for the desired global posterior distribution given the whole dataset. [16] directly combines the samples by weighted averages under an implicit Gaussian assumption; [12] approximates each local posterior with either a Gaussian or a Gaussian kernel density estimate (KDE) so that the combination follows an explicit product of densities; [19] takes the KDE idea one step further by representing it as a Weierstrass transform; [9] uses the "median posterior" in an RKHS embedding space as a combination technique that is robust in the presence of outliers. The main drawback of embarrassingly parallel MCMC sampling is that if the local posteriors differ significantly, perhaps due to noise or non-random partitioning of the dataset across the cluster, or if they do not satisfy the Gaussian as-

---

[*]This work was started and completed when the author was visiting University of Oxford.

sumptions in a number of methods, the final combination stage can result in highly inaccurate global posterior representations.

To encourage local MCMC samplers to roughly be aware of and hence agree with one another so as to improve inference quality, we develop a method to enforce sharing of a small number of moment statistics of the local posteriors, e.g. mean and covariance, across the samplers. We frame our method as expectation propagation (EP) [8], where the exponential family is defined by the shared moments and each node represents a factor to be approximated, with moment statistics to be estimated by the corresponding sampler. Messages passed among the nodes encode differences between the estimated moments, so that at convergence all nodes agree on these moments. As EP tends to converge rapidly, these messages will be passed around only infrequently (relative to the number of MCMC iterations). It can also be performed in an asynchronous fashion, hence incurring low communication costs. As opposed to previous embarrassingly parallel schemes which require a final combination stage, upon convergence each sample drawn at any single node with our method can be directly treated as a sample from an approximate global posterior distribution. Our method differs from standard EP as each factor to be approximated consists of a product of many likelihood terms (rather than just one as in standard EP), and therefore suffers less approximation bias.

## 2 A Distributed Bayesian Posterior Sampling Algorithm

In this section we develop our method for distributed Bayesian posterior sampling. We assume that we have a dataset $\mathcal{D} = \{\mathbf{x}_n\}_{n=1}^N$ with $N \gg 1$ which has already been partitioned onto $m$ compute nodes. Let $\mathcal{D}_i$ denote the data on node $i$ for $i = 1, \ldots, m$ such that $\mathcal{D} = \cup_{i=1}^m \mathcal{D}_i$. Let $\mathcal{D}_{-i} = \mathcal{D} \backslash \mathcal{D}_i$. We assume that the data are i.i.d. given a parameter vector $\boldsymbol{\theta} \in \Theta$ with prior distribution $p_0(\boldsymbol{\theta})$. The object of interest is the posterior distribution, $p(\boldsymbol{\theta}|\mathcal{D}) \propto p_0(\boldsymbol{\theta}) \prod_{i=1}^m p(\mathcal{D}_i|\boldsymbol{\theta})$, where $p(\mathcal{D}_i|\boldsymbol{\theta})$ is a product of likelihood terms, one for each data item in $\mathcal{D}_i$.

Recall that our general approach is to have an independent sampler running on each node targeting a "local posterior", and our aim is for the samplers to agree on the overall shape of the posteriors, by enforcing that they share the same moment statistics, e.g. using the first two moments they will share the same mean and covariance. Let $S(\boldsymbol{\theta})$ be the sufficient statistics function such that $f(S) := \mathbb{E}_f[S(\boldsymbol{\theta})]$ are the moments of interest for some density $f(\boldsymbol{\theta})$. Consider an exponential family of distributions with sufficient statistics $S(\cdot)$ and let $q(\boldsymbol{\theta}; \eta)$ be a density in the family with natural parameter $\eta$. We will assume for simplicity that the prior belongs to the exponential family, $p_0(\boldsymbol{\theta}) = q(\boldsymbol{\theta}; \eta_0)$ for some natural parameter $\eta_0$. Let $\tilde{p}_i(\boldsymbol{\theta}|\mathcal{D}_i)$ denote the *local posterior* at node $i$. Rather than using the same prior, e.g. $p_0(\boldsymbol{\theta})$, at all nodes, we use a *local prior* which enforces the moments to be similar between local posteriors. More precisely, we consider the following target density,

$$\tilde{p}_i(\boldsymbol{\theta}|\mathcal{D}_i) \propto q(\boldsymbol{\theta}; \eta_{-i}) p(\mathcal{D}_i|\boldsymbol{\theta}),$$

where the effective local prior $q(\boldsymbol{\theta}; \eta_{-i})$ is determined by the (natural) parameter $\eta_{-i}$. We set $\eta_{-i}$ such that $\mathbb{E}_{\tilde{p}_i(\boldsymbol{\theta}|\mathcal{D}_i)}[S(\boldsymbol{\theta})] = \mu$ for all $i$, for some shared moment vector $\mu$.

As an aside, note that the overall posterior distribution can be recovered via

$$p(\boldsymbol{\theta}|\mathcal{D}) \propto p(\mathcal{D}|\boldsymbol{\theta})p_0(\boldsymbol{\theta}) = p_0(\boldsymbol{\theta}) \prod_{i=1}^m p(\mathcal{D}_i|\boldsymbol{\theta}) \propto q(\boldsymbol{\theta}; \eta_0) \prod_{i=1}^m \left[ \frac{\tilde{p}_i(\boldsymbol{\theta}|\mathcal{D}_i)}{q(\boldsymbol{\theta}; \eta_{-i})} \right], \qquad (1)$$

for any choice of the parameters $\eta_{-i}$, with a number of previous works corresponding to different choices. [16, 12, 19] use $\eta_{-i} = \eta_0/m$, so that the local prior is $p_0(\boldsymbol{\theta})^{1/m}$ and (1) reduces to $p(\boldsymbol{\theta}|\mathcal{D}) \propto \prod_{i=1}^m \tilde{p}_i(\boldsymbol{\theta}|\mathcal{D}_i)$. [2] set $\eta_{-i} = \eta_0$ for their distributed asynchronous streaming variational algorithm, but reported that setting $\eta_{-i}$ such that $q(\boldsymbol{\theta}; \eta_{-i})$ approximates the posterior distribution given previously processed data achieves better performance. We say that such choice of $\eta_{-i}$ is *context aware* as it contains contextual information from other local posteriors. Finally, in the ideal situation with exact equality, $q(\boldsymbol{\theta}; \eta_{-i}) = p(\boldsymbol{\theta}|\mathcal{D}_{-i})$, then each local posterior is precisely the true posterior $p(\boldsymbol{\theta}|\mathcal{D})$. In the following subsections, we will describe how EP can be used to iteratively approximate $\eta_{-i}$ so that $q(\boldsymbol{\theta}; \eta_{-i})$ matches $p(\boldsymbol{\theta}|\mathcal{D}_{-i})$ as closely as possible in the sense of minimising the KL divergence. Since our algorithm performs distributed sampling by sharing messages containing moment information, we refer to it as **SMS** (in short for *sampling via moment sharing*).

## 2.1 Expectation Propagation

In many typical scenarios the posterior is intractable to compute because the product of likelihoods and the prior is not analytically tractable and approximation schemes, e.g. variational methods or MCMC, are required to compute the posterior. EP is a variational message-passing scheme [8], where each likelihood term is approximated by an exponential family density chosen iteratively to minimise the KL divergence to a "local posterior".

Suppose we wish to approximate (up to normalisation) the likelihood $p(\mathcal{D}_i|\boldsymbol{\theta})$ (as a function of $\boldsymbol{\theta}$), using the exponential family density $q(\boldsymbol{\theta}; \eta_i)$ for some suitably chosen natural parameter $\eta_i$, and that other parameters $\{\eta_j\}_{j\neq i}$ are known such that each $q(\boldsymbol{\theta}; \eta_j)$ approximates the corresponding $p(\mathcal{D}_j|\boldsymbol{\theta})$ well. Then the posterior distribution is well approximated by a local posterior where all but one likelihood factor is approximated,

$$p(\boldsymbol{\theta}|\mathcal{D}) \approx \tilde{p}_i(\boldsymbol{\theta}|\mathcal{D}) \propto p_0(\boldsymbol{\theta})p(\mathcal{D}_i|\boldsymbol{\theta})\prod_{j\neq i}q(\boldsymbol{\theta}; \eta_j) = p(\mathcal{D}_i|\boldsymbol{\theta})\tilde{p}_i(\boldsymbol{\theta}|\mathcal{D}_{-i}),$$

where $\tilde{p}_i(\boldsymbol{\theta}|\mathcal{D}_{-i}) = q(\boldsymbol{\theta}; \eta_{-i})$, with $\eta_{-i} = \eta_0 + \sum_{j\neq i}\eta_j$, is a *context-aware prior* which incorporates information from the other data subsets and is an approximation to the conditional distribution $p(\boldsymbol{\theta}|\mathcal{D}_{-i})$. Replace $p(\mathcal{D}_i|\boldsymbol{\theta})$ by $q(\boldsymbol{\theta}; \eta_i)$, then the corresponding local posterior $\tilde{p}_i(\boldsymbol{\theta}|\mathcal{D})$ would be approximated by $q(\boldsymbol{\theta}; \eta_{-i} + \eta_i)$. A natural choice for the parameter $\eta_i$ is the one that minimises $\mathrm{KL}(\tilde{p}_i(\boldsymbol{\theta}|\mathcal{D})\|q(\boldsymbol{\theta}; \eta_{-i}+\eta_i))$. This optimisation can be solved by calculating the moment parameter $\mu_i = \mathbb{E}_{\tilde{p}_i(\boldsymbol{\theta}|\mathcal{D})}[S(\boldsymbol{\theta})]$, transforming the moment parameter $\mu_i$ into its natural parameter, say $\nu_i$, and then updating $\eta_i \leftarrow \nu_i - \eta_{-i}$.

EP proceeds iteratively, by updating each parameter given the current values of the others using the above procedure until convergence. At convergence (which is not guaranteed), we have that,

$$\nu_i = \nu := \eta_0 + \sum_{j=1}^{m}\eta_j,$$

for all $i$, where $\eta_j$ are the converged parameter values. Hence the natural parameters, as well as the moments of the local posteriors, at all nodes agree. When the prior $p_0(\boldsymbol{\theta})$ does not belong to the exponential family, we may simply treat it as $p(\mathcal{D}_0|\boldsymbol{\theta})$ where $\mathcal{D}_0 = \emptyset$ and approximate it with $q(\boldsymbol{\theta}; \eta_0)$ just as we approximate the likelihoods.

## 2.2 Distributed Sampling via Moment Sharing

In typical EP applications, the moment parameter $\mu_i = \mathbb{E}_{\tilde{p}_i(\boldsymbol{\theta}|\mathcal{D})}[S(\boldsymbol{\theta})]$ can be computed either analytically or using numerical quadrature. In our setting, this is not possible as each likelihood factor $p(\mathcal{D}_i|\boldsymbol{\theta})$ is now a product of many likelihoods with generally no tractable analytic form. Instead we can use MCMC sampling to estimate these moments.

The simplest algorithm involves synchronous EP updates: At each EP iteration, each node $i$ receives from a master node $\eta_{-i}$ (initialised to $\eta_0$ at the first iteration) calculated from the previous iteration, runs MCMC to obtain $T$ samples from which the moments $\mu_i$ are estimated, converts this into natural parameters $\nu_i$, and returns $\eta_i = \nu_i - \eta_{-i}$ to the master node. (Note that the MCMC samplers are run in parallel; hence the moments are computed in parallel unlike standard EP.) An asynchronous version can be implemented as well: At each node $i$, after the MCMC samples are obtained and the new $\eta_i$ parameter computed, the node communicates asynchronously with the master to send $\eta_i$ and receive the new value of $\eta_{-i}$ based on the current $\eta_{j\neq i}$ from other nodes. Finally, a decentralised scheme is also possible: Each node $i$ stores a local copy of all the parameters $\eta_j$ for each $j = 1, \ldots, m$, after the MCMC phase and a new value of $\eta_i$ is computed it is broadcast to all nodes, the local copy is updated based on messages the node received in the mean time, and a new $\eta_{-i}$ is computed.

## 2.3 Multivariate Gaussian Exponential Family

For concreteness, we will describe the required computations of the moments and natural parameters in the special cases of a multivariate Gaussian exponential family. In addition to being analytically tractable and popular, the usage of multivariate Gaussian distribution can also be motivated using

Bayesian asymptotics for large datasets. In particular, for parameters in $\mathbb{R}^d$ and under regularity conditions, if the size of the subset $\mathcal{D}_i$ is large, the Bernstein-von Mises Theorem shows that the local posterior distribution is well approximated by a multivariate Gaussian; hence the EP approximation by an exponential family density will be very good. Given $T$ samples $\{\boldsymbol{\theta}_{it}\}_{t=1}^{T}$ collected at node $i$, unbiased estimates of the moments (mean $\mu_i$ and covariance $\Sigma_i$) are given by

$$\mu_i \leftarrow \frac{1}{T}\sum_{t=1}^{T}\boldsymbol{\theta}_{it} \qquad\qquad \Sigma_i \leftarrow \frac{1}{T-1}\sum_{t=1}^{T}(\boldsymbol{\theta}_{it}-\mu_i)(\boldsymbol{\theta}_{it}-\mu_i)^{\top}, \qquad (2)$$

while the natural parameters can be computed as $\eta_i = (\Omega_i \mu_i, \Omega_i)$, where

$$\Omega_i = \frac{T-d-2}{T-1}\Sigma_i^{-1} \qquad (3)$$

is an unbiased estimate of the precision matrix [11]. Note that simply using $\Sigma_i^{-1}$ leads to a biased estimate, which impacts upon the convergence of EP. Alternative estimators exist [18] but we use the above unbiased estimate for simplicity. We stress that our approach is not limited to multivariate Gaussian, but applicable to any exponential family distribution. In Section 3.2, we consider the case where the local posterior is approximated using the spike and slab distribution.

## 2.4 Additional Comments

The collected samples can be used to form estimates for the global posterior $p(\boldsymbol{\theta}|\mathcal{D})$ in two ways. Firstly, these samples can be combined using a combination technique [16, 12, 19, 9]. According to (1), each sample $\boldsymbol{\theta}$ needs to be assigned a weight of $q(\boldsymbol{\theta}; \eta_{-i})^{-1}$ before being combined. Alternatively, once EP has converged, the MCMC samples target the local posterior $p_i(\boldsymbol{\theta}|\mathcal{D})$, which is already a good approximation to the global posterior, so the samples can be used directly as approximate samples of the global posterior *without need for a combination stage*. This has the advantage of producing $mT$ samples if each of the $m$ nodes produces $T$ samples, while other combination techniques only produce $T$ samples. We have found the second approach to perform well in practice.

In our experiments we have found damping to be essential for the convergence of the algorithm. This is because in addition to the typical convergence issues with EP, our mean parameters are also estimated using MCMC and thus introduces additional stochasticity which can affect the convergence. There is little theory in the literature on convergence of EP [17], and even less can be shown with the additional stochasticity introduced by the MCMC sampling. Nevertheless, we have found that damping the natural parameters $\eta_i$ works well in practice.

In the case of multivariate Gaussians, additional consideration has to be given due to the possibility that the oscillatory behaviour in EP can lead to covariance matrices that are not positive definite. If the precision of a local prior $\Omega_{-i}$ is not positive definite, the resulting local posterior will become unnormalisable and the MCMC sampling will diverge. We adopt a number of mitigating strategies that we have found to be effective: Whenever a new value of the precision matrix $\Omega_{-i}^{\text{new}}$ is not positive definite, we damp it towards its previous value as $\alpha\Omega_{-i}^{\text{old}} + (1-\alpha)\Omega_{-i}^{\text{new}}$, with an $\alpha$ large enough such that the linear combination is positive definite; We collect a large enough number of samples at each MCMC phase to reduce variability of the estimators; And we use the pseudo-inverse instead of actual matrix inverse in (3).

## 3 Experiments

### 3.1 Bayesian Logistic Regression

We tested our sampling via moment sharing method (**SMS**) on Bayesian logistic regression with simulated data. Given a dataset $\mathcal{D} = \{(\mathbf{x}_n, y_n)\}_{n=1}^{N}$ where $\mathbf{x}_n \in \mathbb{R}^d$ and $y_n = \pm 1$, the conditional model of each $y_n$ given $x_n$ is

$$p(y_n|\mathbf{x}_n, \mathbf{w}) = \sigma(y_n\mathbf{w}^{\top}\mathbf{x}_n), \qquad (4)$$

where $\sigma(x) = 1/(1+e^{-x})$ is the standard logistic (sigmoid) function and the weight vector $\mathbf{w} \in \mathbb{R}^d$ is our parameter of interest. For simplicity we did not include the intercept in the model. We used a standard Gaussian prior $p_0(\mathbf{w}) = \mathcal{N}(\mathbf{w}; \mathbf{0}_d, \mathbb{I}_d)$ on $\mathbf{w}$ and the aim is to draw samples from the posterior $p(\mathbf{w}|\mathcal{D})$.

Our simulated dataset consists of $N = 4000$ data points, each with $d = 20$ dimensional covariates, generated using i.i.d. draws $\mathbf{x}_n \sim \mathcal{N}(\boldsymbol{\mu}_x, \Sigma_x)$, where $\Sigma_x = PP^\top$, $P \in [0,1]^{d \times d}$ and each entry of $\boldsymbol{\mu}_x$ and $P$ are in turn generated i.i.d. from $\mathcal{U}(0,1)$. We generate the "true" parameter vector $\mathbf{w}^*$ from the prior $\mathcal{N}(\mathbf{0}_d, \mathbb{I}_d)$, with which the labels are sampled i.i.d. according to the model, i.e. $p(y_n) = \sigma(y_n \mathbf{w}^{*\top} \mathbf{x}_n)$. The dataset is visualized in Fig. 1.

As the base MCMC sampler used across all methods, we used the No-U-Turn sampler (NUTS) [6]. NUTS was also used to generate 100000 samples from the full posterior $p(\boldsymbol{\theta}|\mathcal{D})$ for ground truth. Across all the methods, the sampler was initialised at $\mathbf{0}_d$ and used the first $20d$ samples for burn-in, then thinned every other sample.

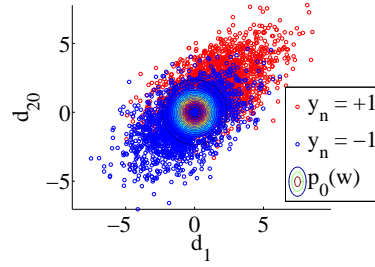

Figure 1: Plot of covariate dimensions 1 and 20 of the simulated dataset for Bayesian logistic regression.

We compared our method **SMS** against consensus Monte Carlo (**SCOT**) [16], the embarrassingly parallel MCMC sampler (**NEIS**) of [12] and the Weierstrass sampler (**WANG**) [19].

**SMS**: We tested both the synchronous (**SMS(s)**) and asynchronous (**SMS(a)**) versions of our method, using a multivariate Gaussian exponential family. The damping factor used was $0.2$. At each EP iteration, **SMS** produced both the EP approximated Gaussian posterior $q(\boldsymbol{\theta}; \eta_0 + \sum_{i=1}^m \eta_i)$, as well as a collection of $mT$ local posterior samples $\Theta$. We use $K$ to denote the total number of EP iterations. For **SMS(a)**, every $m$ worker-master update is counted as one EP iteration.

**SCOT**: Since each node in our algorithm effectively draws $KT$ samples in total, we allowed each node in **SCOT** to draw $KT$ samples as well, using a single NUTS run. To compare against our algorithm at iteration $k \leq K$, we used the first $kT$ samples for combination and form the approximate posterior samples.

**NEIS**: As in **SCOT**, we drew $KT$ samples at each node, and compared against ours at iteration $k$ using the first $kT$ samples. We tested both the parametric (**NEIS(p)**) and non-parametric (**NEIS(n)**) combination methods. To combine the kernel density estimates in **NEIS(n)**, we adopted the recursive pairwise combination strategy as suggested in [12, 19]. We retained $10mT$ samples during intermediate stages of pair reduction and finally drew $mT$ samples from the final reduction.

**WANG**: We test the sequential sampler in the first arXiv version, which can handle moderately high dimensional data and does not require a good initial approximation. The bandwidths $h_l$ $(l = 1, \ldots, d)$ were initialized to $0.01$ and updated with $\sqrt{m}\sigma_l$ (if smaller) as suggested by the authors, where $\sigma_l$ is the estimated posterior standard deviation of dimension $l$. As a Gibbs sampling algorithm, **WANG** requires a larger number of iterations for convergence but does not need as many samples within each iteration. Hence we ran it for $K' = 700 \gg K$ iterations, each time generating $KT/K'$ samples on every node. We then collected every $T$ combined samples generated from each subsequent $K'/K$ iterations for comparative purposes, leaving all previous samples as burn-in.

All methods were implemented and tested in Matlab. Experiments were conducted on a cluster with as many as 24 nodes (Matlab workers), arranged in 4 servers, each being a multi-core server with 2 Intel(R) Xeon(R) E5645 CPUs (6 cores, 12 threads). We used the `parfor` command (synchronous) and the `parallel.FevalFuture` object (asynchronous) in Matlab for parallel computations. The underlying message passing is managed by the Matlab Distributed Computing Server.

**Convergence of Shared Moments**. Figure 2 demonstrates the convergence of the local posterior means as the EP iteration progresses, on a smaller dataset generated likewise with $N = 1000$, $d = 5$ and 25000 samples as ground truth. It clearly illustrates that our algorithm achieves very good approximation accuracy by quickly enforcing agreement across nodes on local posterior moments (mean in this case). When $m = 50$, we used a larger number of samples for stable convergence.

**Approximation Accuracies**. We compare the approximation accuracy of the different methods on our main simulated data ($N = 4000$, $d = 20$). We use a moderately large number of nodes $m = 32$, and $T = 10000$. In this case, each subset consists of 125 data points. We considered three different error measures for the approximation accuracies. Denote the ground truth posterior samples, mean and covariance by $\Theta^*$, $\boldsymbol{\mu}^*$ and $\Sigma^*$, and correspondingly $\widehat{\Theta}$, $\widehat{\boldsymbol{\mu}}$ and $\widehat{\Sigma}$ for the approximate samples collected using a distributed MCMC method. The first error measure is mean squared error (MSE)

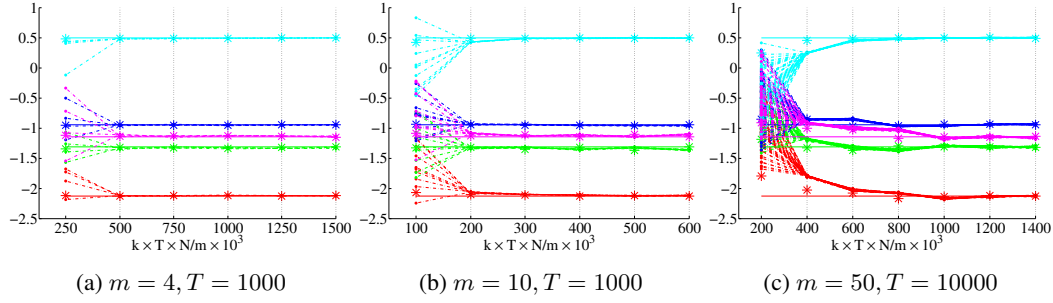

(a) $m = 4, T = 1000$    (b) $m = 10, T = 1000$    (c) $m = 50, T = 10000$

Figure 2: Convergence of local posterior means on a smaller Bayesian logistic regression dataset ($N = 1000, d = 5$). The x-axis indicates the number of likelihood evaluations, with vertical lines denoted EP iteration numbers. The y-axis indicates the estimated posterior means (dimensions indicated by different colours). We show ground truth with solid horizontal lines, the EP estimated mean with asterisks, and local sample estimated means dots connected with dash lines.

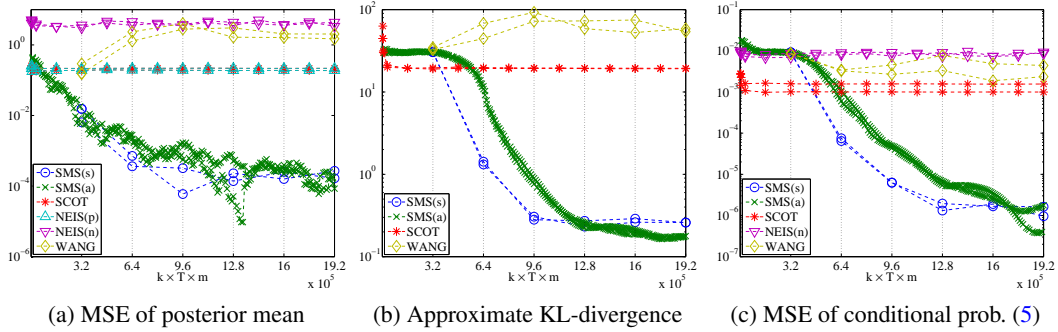

(a) MSE of posterior mean    (b) Approximate KL-divergence    (c) MSE of conditional prob. (5)

Figure 3: Errors (log-scale) against the cumulative number of samples drawn on all nodes ($kTm$). We tested two random splits of the dataset (hence 2 curves for each algorithm). Each complete EP iteration is highlighted by a vertical grid line. Note that for **SCOT**, **NEIS(p)** and **NEIS(n)**, apart from usual combinations that occur after every $Tm/2$ local samples are drawn on all nodes, we also deliberately looked into combinations at a much earlier stage at $(0.01, 0.02, 0.1, 0.5)Tm$.

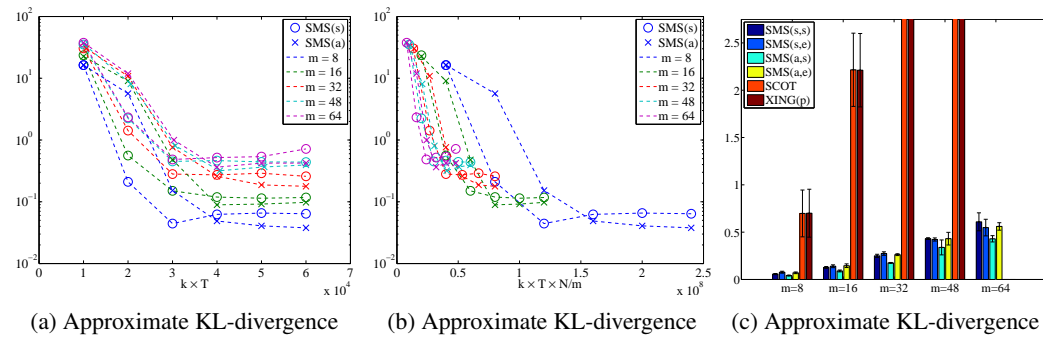

(a) Approximate KL-divergence    (b) Approximate KL-divergence    (c) Approximate KL-divergence

Figure 4: Cross comparison with different numbers of nodes. Note that the x-axes have different meanings. In figure (a), it is the cumulative number of samples drawn locally on each node ($kT$). For the asynchronous **SMS(a)**, we only plot every $m$ iterations so as to mimic the behaviour of **SMS(s)** for a more direct comparison. In figure (b) however, it is the cumulative number of likelihood evaluations on each node ($kTN/m$), which more accurately reflect computation time.

between $\widehat{\boldsymbol{\mu}}$ and $\boldsymbol{\mu}^*$: $\sum_{l=1}^{d}(\widehat{\boldsymbol{\mu}}_l - \boldsymbol{\mu}_l^*)^2/d$; the second is KL-divergence between $\mathcal{N}(\boldsymbol{\mu}^*, \Sigma^*)$ and $\mathcal{N}(\widehat{\boldsymbol{\mu}}, \widehat{\Sigma})$; and finally the MSE of the conditional probabilities:

$$\frac{1}{N} \sum_{\mathbf{x} \in \mathcal{D}} \Big[ \frac{1}{|\widehat{\Theta}|} \sum_{\mathbf{w} \in \widehat{\Theta}} \sigma(\mathbf{w}^\top \mathbf{x}) - \frac{1}{|\Theta^*|} \sum_{\mathbf{w} \in \Theta^*} \sigma(\mathbf{w}^\top \mathbf{x}) \Big]^2. \tag{5}$$

Figure 3 shows the results for two separate runs of each method. We observe that both versions of **SMS** converge rapidly, requiring few rounds of EP iterations. Further, they produce approximation errors significantly below other methods. The synchronous **SMS(s)** does appear more stable and converges faster than its asynchronous counterpart but ultimately both versions achieve the same level of accuracy. **SCOT** and **NEIS(p)** are very closely related, with their MSE for posterior mean overlapping. Both methods achieve reasonable accuracy early on, but fail to further improve with the increasing number of samples available for combination due to their assumptions of Gaussianity. **NEIS(p)** directly estimates $\widehat{\boldsymbol{\mu}}$ and $\widehat{\Sigma}$ without drawing samples $\widehat{\Theta}$ and is thus missing from Figure 3b and 3c. Note that **NEIS(n)** is missing from Figure 3b because the posterior covariance estimated from the combined samples is singular due to an insufficient number of distinct samples. Unsurprisingly, **WANG** requires a large number of iterations for convergence and does not achieve very good approximation accuracy. It is also possible that the poor performances of **NEIS(n)** and **WANG** are due to the kernel density estimation used, as its quality deteriorates very quickly with dimensionality.

**Influence of the Number of Nodes**. We also investigated how the methods behave with varying numbers of partitions, $m = 8, 16, 32, 48, 64$. We tested the methods on three runs with three different random partitions of the dataset. We only tested $m = 64$ on our **SMS** methods.

In Figure 4a, we see the rapid convergence in terms of the number of EP iterations, and the insensitivity to the number of nodes. Also, the final accuracies of the **SMS** methods are better for smaller values of $m$. This is not surprising since the approximation error of EP tends to increase when the posterior is factorised into more factors. In the extreme case of $m = 1$, the methods will be exact. Note however that with larger $m$, each node contains a smaller subset of data, and computation time is hence reduced. In Figure 4b we plotted the same curves against the number $kTN/m$ of likelihood evaluations on each node, which better reflects the computation times. We thus see an accuracy-computation time trade-off, where with larger $m$ computation time is reduced but accuracies get worse. In Figure 4c, we looked into the accuracy of the obtained approximate posterior in terms of KL-divergence. Note that apart from a direct read-off of the mean and covariance from the parametric EP estimate (**SMS(s,e)** & **SMS(a,e)**), we might also compute the estimators from the posterior samples (**SMS(s,s)** & **SMS(a,s)**), and we compared both of these in the figure. As noted above, the accuracies are better when we have less nodes. However, the errors of our methods still increase much slower than **SCOT** and **NEIS(p)**, for both of which the KL-divergence increases to around 20 and 85 when $m = 32$ and $48$ and is thus cropped from the figure.

## 3.2 Bayesian sparse linear regression with spike and slab prior

In this experiment, we apply **SMS** to a Bayesian sparse linear regression model with a spike and slab prior over the weights. Our goal is to illustrate that our framework is applicable in scenarios where the local posterior distribution is approximated by other exponential family distributions and not just the multivariate Gaussian.

Given a feature vector $\mathbf{x}_n \in \mathbb{R}^d$, we model the label as $y_n \sim \mathcal{N}(\mathbf{w}^\top \mathbf{x}_n, \sigma_y^2)$, where $\mathbf{w}$ is the parameter of interest. We use a spike and slab prior [10] over $\mathbf{w}$, which is equivalent to setting $\mathbf{w} = \widetilde{\mathbf{w}} \odot \boldsymbol{s}$, where $\boldsymbol{s}$ is a $d$-dimensional binary vector (where $1$ corresponds to an active feature and $0$ inactive) whose elements are drawn independently from a Bernoulli distribution whose natural (log odds) parameter is $\beta_0$ and $\widetilde{w}_l | s_l \sim \mathcal{N}(0, \sigma_w^2)$ i.i.d. for each $l = 1, \ldots, d$. [7] proposed the following variational approximation of the posterior: $q(\widetilde{\mathbf{w}}, \boldsymbol{s}) = \prod_{l=1}^{d} q(\widetilde{w}_l, s_l)$ where each factor $q(\widetilde{w}_l, s_l) = q(s_l)q(\widetilde{w}_l | s_l)$ is a spike and slab distribution. (We refer the reader to [7] for details.)

The spike and slab distribution over $\boldsymbol{\theta} = (\widetilde{\mathbf{w}}, \boldsymbol{s})$ is an exponential family distribution with sufficient statistics $\{s_l, s_l \widetilde{w}_l, s_l \widetilde{w}_l^2\}_{l=1}^{d}$, which we use for the EP approximation. The moments required consist of the probability of $s_l = 1$, and the mean and variance of $\widetilde{w}_l$ conditioned on $s_l = 1$, for each $l = 1, \ldots, d$. The conditional distribution of $\widetilde{w}_l$ given $s_l = 0$ is simply the prior $\mathcal{N}(0, \sigma_w^2)$. The natural parameters consist of the log odds of $s_l = 1$, as well as those for $\widetilde{w}_l$ conditioned on $s_l = 1$

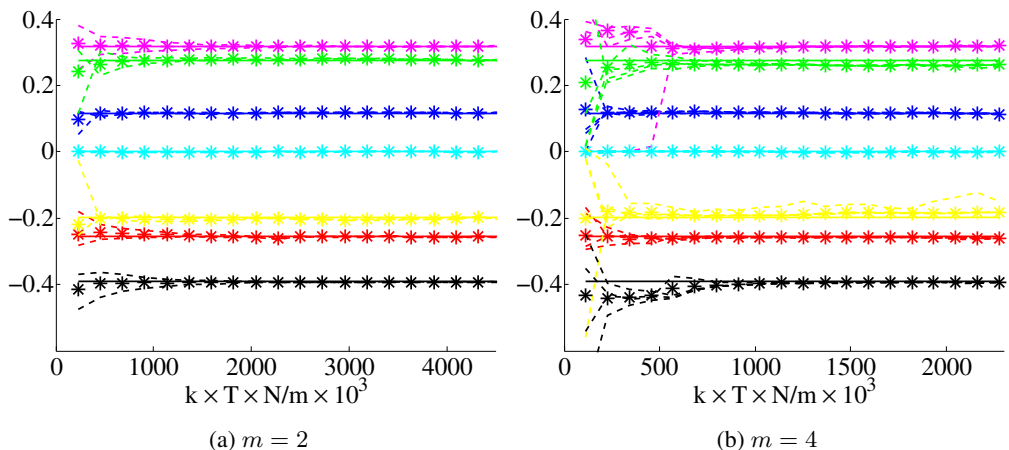

(a) $m = 2$           (b) $m = 4$

Figure 5: Results on Boston housing dataset for Bayesian sparse linear regression model with spike and slab prior. The x-axis plots the number of data points per node (equals the number of likelihood evaluations per sample) times the cumulative number of samples drawn per node, which is a surrogate for the computation times of the methods. The y-axis plots the ground truth (solid), local sample estimated means (dashed) and EP estimated mean (asterisks) at every iteration.

(Section 2.3). We used the paired Gibbs sampler described in [7] as the underlying MCMC sampler, and a damping factor of $0.5$.

We experimented using the Boston housing dataset which consists of $N = 455$ training data points in $d = 13$ dimensions. We fixed the hyperparameters to the values described in [7], and generated ground truth samples by running a long chain of the paired Gibbs sampler and computed the posterior mean of $\mathbf{w}$ using these ground truth samples. Figure 5 illustrates the output of **SMS(s)** for $m = 2$ and $m = 4$ (the number of nodes was kept small to ensure that each node contains at least 100 observations). Each color denotes a different dimension; to avoid clutter, we report results only for dimensions $2, 5, 6, 7, 9, 10,$ and $13$. The dashed lines denote the local sample estimated means at each of the nodes; the solid lines denote the ground truth and the asterisks denote the EP estimated mean at each iteration. Initially, the local estimated means are quite different since each node has a different random data subset. As EP progresses, these local estimated means as well as the EP estimated mean converge rapidly to the ground truth values.

## 4 Conclusion

We proposed an approach to performing distributed Bayesian posterior sampling where each compute node contains a different subset of data. We show that through very low-cost and rapidly converging EP messages passed among the nodes, the local MCMC samplers can be made to share a number of moment statistics like the mean and covariance. This in turn allows the local MCMC samplers to converge to the same part of the parameter space, and allows each local sample produced to be interpreted as an approximate global sample without the need for a combination stage. Through empirical studies, we showed that our methods are more accurate than previous methods and also exhibits better scalability to the number of nodes. Interesting avenues of research include using our **SMS** methods to adjust hyperparameters using either empirical or fully Bayesian learning, implementation and evaluation of the decentralised version of **SMS**, and theoretical analysis of the behaviour of EP under the stochastic perturbations caused by the MCMC estimation of moments.

**Acknowledgements**

We thank Willie Neiswanger for sharing his implementation of **NEIS(n)**, and Michalis K Titsias for sharing the code used in [7]. MX, JZ and BZ gratefully acknowledge funding from the National Basic Research Program of China (No. 2013CB329403) and National NSF of China (Nos. 61322308, 61332007). BL gratefully acknowledges generous funding from the Gatsby charitable foundation. YWT gratefully acknowledges EPSRC for research funding through grant EP/K009362/1.

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
