[Reviews · NeurIPS 2014]

Submitted by Assigned_Reviewer_4

This paper proposes a new distributed Bayesian Posterior inference algorithm for big data, an important problem that has garnered a lot of attention in the last few years. The goal is to sample from the posterior distribution of model parameters theta given a dataset that is divided among m machines. The proposed distributed algorithm runs a separate Markov chain on each machine, each of which samples from a distribution proportional to p(Dm|theta) qm(theta) where Dm is the local data subset on machine m and qm is a variational (Gaussian) approximation to the product of similar factors on other machines. qm is adapted over time by passing low cost messages between machines.

I think this is a great idea and addresses a very important problem. The paper is also relatively easy to understand. However, the experimental section is quite weak which makes me a little hesitant to argue strongly for acceptance. At present, the authors evaluate the algorithm only on a Bayesian logistic regression problem with a small synthetic dataset (N = 4K or 10K and D =20). This is OK as a proof of concept, but some more experiments can greatly improve the paper, e.g.:

1) The different algorithms are compared in terms of how they reduce error vs number of samples, but I would also like to see a plot of error vs computational time in seconds. The proposed algorithm does seem efficient, but it would still be nice to see this in an actual plot. I would also like to see a comparison (in terms of error vs time in secs) with a non-distributed algorithm (e.g. the NUTS sampler mentioned in the paper) running on a single machine using all the data. This is to show that the communication costs and reduced mixing from not using all the data is compensated for by the reduction in computational cost per sample.

2) On a similar note, It would also be interesting to see plots of error vs time, for different number of partitions. The error seems to increase with more partitions, but increasing the partitions could potentially speed up computation (if communication costs are not too high).

3) The algorithm is most useful for big datasets that cannot fit on a single machine, so I think it is necessary that the paper have some experiments on actual big datasets. Also, if each machine itself has a lot of data, one can use this algorithm with a mini-batch based approximate MCMC algorithm (see reference [1] below and follow up papers that cite this) as the local sampler on each machine instead of the NUTS sampler.

4) The paper uses the Bernstein von-Mises theorem to motivate the use of Gaussian approximations to the posterior and sub-factors. It is fine to derive the algorithm like this, but I think it is still necessary to empirically test how the performance degrades for significantly non-Gaussian looking posteriors because this is the case where we actually want to use an MCMC algorithm. If the posterior was very close to Gaussian, one could get away with simple techniques such as Laplace approximation.

5) It would greatly strengthen the paper to have an experiment on a problem where performing full Bayesian inference using an expensive procedure such as MCMC is actually worthwhile. By this, I mean a problem where full Bayesian inference can be shown to be better than a simple straw man, such as MAP estimation or Laplace approximation, in terms of prediction accuracy, avoiding over-fitting or providing useful uncertainty levels.

References:
1. Stochastic Bayesian Learning via Stochastic Gradient Langevin Dynamics - Welling and Teh, 2011
Summary: A great idea addressing an important problem, but I feel it is not ready yet and needs some more work. Better experiments will greatly improve the paper.

Submitted by Assigned_Reviewer_12

Summary: The authors propose a new Bayesian inference algorithm that operates on subsets of data in a distributed environment. This work is in the spirit of an active area of recent work on parallel MCMC for Bayesian inference using subsets of data. The authors focus on comparisons to parallel methods; a complementary line of work uses data subsets as mini-batches. (I think these are not discussed, though they could be.) The new method, eventually called Distributed Context-Aware Sampling (DCAS), combines MCMC and expectation propagation. All these methods must eventually share information between parallel cores, each of which "owns" a data subset. The strategy here is to share some information between nodes, encoded in a "context prior." In the experimental evaluation, the proposed method looks better than some other recent methods, but the experiments are very limited.

Quality: The proposed method is interesting and well-motivated. Theoretical results would have been nice, but not necessarily required -- in which case I would expect thorough and impressive experiments. The algorithm relies on Normal models, fitted at each node, to share information between nodes. A major question I have is, in the case where everything is actually Gaussian, does the algorithm converge to the exact target?
The writing is mostly good, except in the Intro, which appears to have been slapped together -- after a short but clear Abstract, the Intro begins immediately with a series of grammatically incoherent sentences containing embarrassing errors. The citations, especially in the first paragraph, are dropped in and lack context (ironically). Please explain the differences between these citations, and point out which concern Bayesian posterior sampling and which do not.
The Experiments are designed thoughtfully, and effort has been made to present a fair comparison of different sampling procedures, where it isn't obvious how to compare them directly. The synthetic problem is nice (Bayesian logistic regression, 4,000 and 10,000 data points, 20 dimensions), but the evaluation could have benefitted from studying several problems along several axes -- synthetic and real data, larger datasets (which are free for synthetic models). I would have liked to have seen standard MCMC convergence tests.

Clarity: Since you are introducing the notion of the context prior, please italicize it in the Abstract. It would also be useful to give some intuition there for what it means and why it is useful. Also in your Abstract, you could elaborate on your results. "We demonstrate the advantages of our algorithm" -- With respect to what? Measured how? Using how many cores? What would be a summary of your quantitative results? Your Abstract and Intro should introduce the name of your algorithm (DCAS), especially since it is not in your title.
Please define acronyms such as KDE and RKHS in the Intro.
The context prior is a nice idea and I appreciated its motivation with respect to the SVB paper. In lines 068-069 where you say, "We believe ... it appears intuitive to suggest ... " -- Can you make stronger statements? Or just explain that you are taking a reasonable Bayesian approach.
There is a lot of notation -- p's and q's with various superscripts and subscripts. I wonder if it can be made clearer or more intuitive. Perhaps there could be a little table, even as an appendix, summarizing all notation. The superscripts (I think they are chi and iota) are not terribly intuitive -- or is there an explanation? chi because it looks like an "x" for conteXt? It took me a while to internalize that the subscript notation refers to a data subset and is also used to suppress this explicit dependence in the usual probabilistic notation -- e.g., p_i(theta) replaces p(theta | D_i). I found this confusing on my initial reading because this led me to interpret the notation for subset posteriors as priors.
Footnote 2 appears at the end of a mathematical expression and looks like an exponent.
Thank you for the sentence around line 087, "To recap ..."
Please define "proj" in Eq. (7).
Section 2.4 could use some high-level sentences at the beginning, explaining the layout of the section.
In 2.4.1, line 132, I think you are defining p_i(theta) here, so please make this clear, as I was searching for it in the previous notation. Also, calling this the "target distribution" may be confusing, since it isn't the true target.
Question about Algorithm 1: should the 1/(T_i - 1) terms be 1/T_i?
Line 161, "As the underlying message passing progresses" -- this is confusing, as you haven't yet described the message passing component of your algorithm, which comes in the next section, 2.4.2. This would be addressed by giving a high-level overview at the beginning of Section 2.4.
Some of the text in Figure 1 is way too small.
The descriptions of the synchronous, asynchronous and decentralized algorithms are very concise but could be expanded somewhat for clarity. What explicit parallel architecture do you have in mind? Who is receiving the messages? Is there a master node? It seemed that the implementation uses shared memory, available to all nodes, to share/update global information. The description of the asynchronous scheme should make explicit how the latest historical copy of q_i^iota is used, using the notation introduced in line 185.
2.4.4, line 226 -- Where did tilde{f} come from?
Line 230, "Several possible remedies have been proposed" -- by whom? You?
Experiments section -- please give explicit details about the cluster architecture -- you mention 8 nodes, is it 1 core per node? It sounds like there is shared memory? I assume then there is no need for MPI? There are 8 nodes, but you study m = 8, 16, 24, 36, 48 partitions -- this suggests to me you have multiple cores per node. The compute architecture needs to be clarified.
Make explicit that you do not evaluate the decentralized scheme.
It would be useful to explicitly describe the combining rules you use for SCOT, NEIS and WANG; this could go in Supplemental. For WANG, you use their sequential sampler -- the corresponding paper proposes at least 3 sampling procedures, why did you choose this one? What might you have expected to be different if you had used one of their other procedures?
Line 261, "Upon convergence" -- according to what criteria?
Please label the axes of Figures 3 and 5 (# samples).
Please make explicit how the ground truth samples are used, and precisely what KL-divergence is measuring (presumably between distributions fit to the ground truth samples and to the various MCMC samples.
If you feel you would benefit from more room, I would suggest moving (parts of) section 3.4 to Supplemental.

Originality: To the best of my knowledge, the proposed method is novel. This work has the same motivations as recent parallel, approximate MCMC schemes that operate on subsets (partitions) of data. These other schemes either sample in an embarrassingly parallel manner, and then need to "fix up" the samples by combining them somehow, or have each parallel core sample from an approximate target (Weierstrass sampling). In all these methods, information must eventually be shared between nodes. This work uses the novel idea of the context prior, which is continuously updated during execution.

Significance: There is much recent interest in the area of scaling MCMC sampling by subsampling (partitioning) the data, and having separate parallel cores process each subsample. Therefore the proposed algorithm is of interest and timely. However, the paper presents no theoretical analysis or guarantees and therefore relies on empirical studies. The empirical evaluation presented here compares with several other recent techniques, but is limited. I really wanted to see a more enlightened discussion of this method compared to other (recent) work -- several other of these methods rely on Normal models, and a discussion/comparison would be very useful.

Other: At least 4 of your references cite arxiv versions when (sometimes recent) journal or conference versions are now available [2, 4, 8, 15].
Line 067, "prompts our core" (delete "us to")
Line 081, commas around "e.g."
Typo around line 090, ", And ..."
Line 095 "two" not "2"
Line 103, "progress"
Line 104, "rules (4) naturally reduce"
Line 110, "first" not "firstly"
Line 215, explicitly define your notation for the zero matrix
Line 226, "not much of"
Line 266, "draw" not "draws"
Summary: The paper presents a novel method in an area of much recent interest (parallel MCMC using data subsets). However, the overall paper is weak because there are no theoretical results, very limited experiments and does not provide an enlightened discussion/critique of recent methods.

Submitted by Assigned_Reviewer_23

This paper develops an approach for MCMC for large data. It involves using using different processors to run MCMC on different subsets of the data. The novelty is in updating the prior used for each subset over time (to approximate the posterior given the data not being analysed by that subset).

This is an important area. And new ideas in this area have potential for high impact. There is clear, albeit limited, novelty in the paper.

The paper is generally clear, though the written English could be improved substantially. [E.g. the first paragraph of the intro should probably read something like

"As The size of data grows to ever increasing scales, recent years have seen a surging need for machine learning algorithms that can scale to large data applications. One, and arguably the most direct, way to scale up these algorithms is to resort to distributed algorithms, and various approaches have been proposed to this end [11, 10, 1]. One speciļ¬c line of work in this area concentrates on applying the popular “divide-and-conquer” idea [15, 4]."

Also some details are missing (e.g. defining the natural parameterisation of the Normal; formalling defining the mathematical object "proj"; stating the p/q only exists if Sigma_1 > Sigma_2). It is also unclear what the output of your approach is (do you pool the samples from the individual MCMC runs? Or do you approximate the posterior using the final EP approximation). Also you list a lot of strategies in 2.4.4 -- which do you use in practice and when?

The simulation results are promising -- though this is a very simple example (where Gaussian approximations are likely to be very accurate).

I was surprised not to see more on the computational cost of the various algorithms.
Summary: An interesting, if slightly incremental idea, to tackle a very important problem. A brief empirical study suggests this approach has promise.
Author Feedback
Author rebuttal: We thank all the reviewers for their acknowledgements of our novel contributions and the very helpful comments and suggestions to thoroughly improve our presentation clarity and experiments. Below we first address common concerns and then respond to individual ones.

* Common concerns:
1. Experiments.
We acknowledge that our reported experiments are limited and inadequate to showcase the full potential of our method. Our main focus back then is to first empirically verify its core properties, namely improvement of accuracy, fast convergence, asynchronous updates, etc. We will include more results in the revision.

2. Generality of the proposed approach.
Our method is actually fairly general and can be easily extended to support more complex non-Gaussian approximations. E.g., the approximating family from which q_i takes form can actually be any member of the exponential family; arguably our approach can be applied to almost all the problems EP is applicable to.

* AR_12:
Thanks for carefully pointing out typos and clarity issues (e.g., in the Intro). We will thoroughly address them in the revision.

1. Discussion of methods using mini-batches.
Indeed stochastic methods based on mini-batches have been another popular line of works to tackle big data problems and it would be interesting to include them into discussion. However it might be noteworthy that those methods, until quite recently ([1]), are not viewed as parallel methods which are the main target we are comparing against in this paper. We will discuss them in the revision.

[1] S. Ahn, B. Shahbaba & M. Welling, Distributed Stochastic Gradient MCMC, ICML’14

2. Theoretical results.
To the best of our knowledge, little is known in general about the convergence properties of even standard EP (e.g., Theorem 1 in [2] at best shows that EP is numerically stable for log-concave factor functions). Such analyses seem even harder in our case since we introduced another level of stochastic approximation (the MCMC-PROJ step) into the algorithm. Nonetheless, we are still studying our algorithm (just as people are still doing EP) and we will keep an eye on such results.

[2] M. W. Seeger, Bayesian Inference and Optimal Design for the Sparse Linear Model, P. 759-813, JMLR’08

3. In case everything is actually Gaussian…
Standard Gaussian-EP converges to the true posterior in that case after just the 1st round. According to our tests, as long as we obtain enough samples during MCMC-PROJ to restrain stochastic error within a limited level, our algorithm also nicely converges to the exact target within very few rounds of iterations.

4. Intuition for the “context prior”.
Yes we agree that we could have argued more strongly and clearly that Eq.2 follows a “reasonable Bayesian approach”.

5. Notations.
We will reconsider our notations to make them clear and intuitive.

6. Denominator: T_i – 1 or T_i?
We chose to divide by (T_i - 1) since it gives an unbiased estimate to the covariance matrix (under our Gaussian assumption).

7. Description of different message passing schemes.
In both the synchronous and asynchronous schemes, there is a master node where the global shared memory resides. But it is no longer needed in the decentralized scheme, which is presumably more robust to node failures. We have not tested it yet but we think including it into discussion helps demonstrating the potential of our method.
We will move the expression in Figure 1(b) to the text for better readability.

8. tilde{f}_i.
We are sorry for the inconsistency due to a historical version. We’ll replace it with q^iota_i in the revision.

9. Architecture.
We used a multi-core computer server with 2 Intel E5-2440 CPUs (12 cores, 24 threads). For now, we are using the “parfor” command (synchronous) and the “parallel.FevalFuture” object (asynchronous) from Matlab for parallel computations. The underlying message passing is mainly managed by Matlab.

10. Choice of WANG’s sampler.
We choose its sequential sampler because it seems to be the only one that can handle moderate (d=20) dimensional dataset and does not require a good initial approximation, which matches our set up.

11. Convergence criterion.
We described it in Line 242~244 and will make it more explicit.

* AR_23:
1. Strategies in 2.4.4.
We used all the four strategies we have listed.

2. Output of the algorithm.
Our algorithm outputs both the approximate Gaussian posterior and a collection of samples generated from its last iteration (Line 261~262).

3. Computational cost.
This involves both the cost for local sampling and that for sample combination. The cost for local sampling is almost identical across the algorithms given that the bottleneck resides in the evaluation of the subset likelihoods. On the other hand, our algorithm is free from sample combination while SCOT, NEIS(p) and WANG all have similar minor cost due to either sample averaging (SCOT & WANG) or statistics summation (NEIS(p)). NEIS(N) has a slightly more complicated combination stage which resamples from the (T^m)-component Gaussian mixture model.

* AR_4:
We understand that measuring the actual running time of different algorithms gives a more direct interpretation of their efficiencies yet we carefully chose our current x-axis to be “# of samples” mainly for two reasons. Firstly, we find that across all the algorithms in study, local sampling actually accounts for a dominant proportion of the entire running time. Secondly, due to practical reasons we ran our experiments with 8 nodes (or workers actually, please refer to our response 10 to AR_12) on a heavily-loaded computer server shared with many other users and hence credible timing information may not be easy to obtain, especially when we have more than 8 partitions. For now, we are considering a reimplementation and deploying it onto a more suitable computation platform for further studies (bigger datasets, more partitions, etc.)